# Application of a Machine Learning Technology in the Definition of Metabolically Healthy and Unhealthy Status: A Retrospective Study of 2567 Subjects Suffering from Obesity with or without Metabolic Syndrome

**DOI:** 10.3390/nu14020373

**Published:** 2022-01-15

**Authors:** Davide Masi, Renata Risi, Filippo Biagi, Daniel Vasquez Barahona, Mikiko Watanabe, Rita Zilich, Gabriele Gabrielli, Pierluigi Santin, Stefania Mariani, Carla Lubrano, Lucio Gnessi

**Affiliations:** 1Department of Experimental Medicine, Section of Medical Pathophysiology, Food Science and Endocrinology, Sapienza University of Rome, 00161 Rome, Italy; renata.risi@uniroma1.it (R.R.); biagi.1778697@studenti.uniroma1.it (F.B.); vasquezbarahona.1771837@studenti.uniroma1.it (D.V.B.); mikiko.watanabe@uniroma1.it (M.W.); s.mariani@uniroma1.it (S.M.); carla.lubrano@uniroma1.it (C.L.); lucio.gnessi@uniroma1.it (L.G.); 2MRC Metabolic Diseases Unit, MRC Institute of Metabolic Science, University of Cambridge, Cambridge CB2 1TN, UK; 3Mix-x Partner, 20153 Milano, Italy; rita.zilich@mix-x.com; 4Rulex Inc., 16122 Genova, Italy; g.gabrielli@rulex.ai; 5Deimos Engineering, 33100 Udine, Italy; p.santin@e-deimos.it

**Keywords:** metabolic syndrome, insulin-like growth factor 1, artificial intelligence

## Abstract

The key factors playing a role in the pathogenesis of metabolic alterations observed in many patients with obesity have not been fully characterized. Their identification is crucial, and it would represent a fundamental step towards better management of this urgent public health issue. This aim could be accomplished by exploiting the potential of machine learning (ML) technology. In a single-centre study (*n* = 2567), we used an ML analysis to cluster patients with metabolically healthy (MHO) or metabolically unhealthy (MUO) obesity, based on several clinical and biochemical variables. The first model provided by ML was able to predict the presence/absence of MHO with an accuracy of 66.67% and 72.15%, respectively, and included the following parameters: HOMA-IR, upper body fat/lower body fat, glycosylated haemoglobin, red blood cells, age, alanine aminotransferase, uric acid, white blood cells, insulin-like growth factor 1 (IGF-1) and gamma-glutamyl transferase. For each of these parameters, ML provided threshold values identifying either MUO or MHO. A second model including IGF-1 zSDS, a surrogate marker of IGF-1 normalized by age and sex, was even more accurate with a 71.84% and 72.3% precision, respectively. Our results demonstrated high IGF-1 levels in MHO patients, thus highlighting a possible role of IGF-1 as a novel metabolic health parameter to effectively predict the development of MUO using ML technology.

## 1. Introduction

Artificial intelligence (AI) is becoming increasingly present in the swiftly evolving medical field, and it is expected to generate impactful advancements in the management of a variety of diseases. The potential medical applications of AI are endless and include the possibility of focusing on primary or secondary prevention, personalisation of treatment, evaluation of risk factors and likelihood of developing specific disorders. Machine learning (ML) is a form of AI which creates algorithms, learning from and acting on data [1]. Unlike traditional analytical approaches, ML can probe information even with only a small amount of prior knowledge and learning from data given as input [2]. The advantage of ML is the possibility to analyse an increasing amount of qualitative and quantitative data in an integrated system [3]. ML has already been successfully exploited to design the best model to yield good metabolic control in type 2 diabetes mellitus (T2DM) [2] and to predict the risk of obesity in early childhood and young people [4,5]. In certain diseases such as obesity, marked by a wide variety of phenotypes and heterogenous manifestations, ML has the potential to optimally characterise individuals, and can provide valuable information to design a personalised management plan. With the help of ML technology, a recent study has succeeded in subclassifying obese phenotypes into different metabolic clusters, reflecting underlying pathophysiology [6].

Obesity is defined as an abnormal fat accumulation, with a detrimental effect on health that has been historically diagnosed as a body mass index (BMI) equal or greater than 30 kg/m^2^ [7,8]. The current diagnostic criteria, however, have poorly characterized the obese population, as they do not take into account body fat distribution, which is largely responsible for the cardiometabolic risk associated with obesity. The pattern of fat deposition presents with a great interindividual variability and results in different clinical presentations. As an example, visceral fat has been associated with a growing burden of noncommunicable diseases, such as metabolic syndrome, diabetes and cardiovascular disease [9]. The metabolic syndrome refers to the co-occurrence of several known cardiovascular risk factors, including altered glucose metabolism, obesity, atherogenic dyslipidaemia and hypertension. There has been recent controversy about its definition, although the most widely used criteria for the diagnosis are those established by the National Cholesterol Education Program Adult Treatment Panel III (NCEP ATP III) and the International Diabetes Federation (IDF) [9]. Given the frequent association between metabolic syndrome and obesity, clinical scientists distinguish a metabolically healthy obesity (MHO), characterized by the absence of the parameters defining metabolic syndrome except for waist circumference, from a metabolically unhealthy obesity (MUO), characterized by a significantly higher risk of complications and mortality [10]. The factors involved in the pathogenesis of metabolic impairment in obesity have yet to be fully elucidated. As far as cardiovascular risk is concerned, the prognostic significance of obesity phenotypes is still under debate; a few studies have characterised their transition trajectories considering that alterations in the physical activity level and morbidity disabilities may precede the onset of metabolic abnormalities [11]. Findings from epidemiological studies have shown that the prevalence of MHO ranges from less than 10% to almost 50% in obese individuals according to different definitions of metabolic health and the population studied [12,13,14]. Substantially, poor metabolic health may increase mortality regardless of obesity status [15,16]. The characterization of metabolic status would allow to identify obese patients who are at higher risk of complications, since moderate weight loss can be sufficient to transition from MUO to MHO and might also lower the risk of adverse outcomes. Applying the concept of metabolic health in management strategies may allow to easily achieve attainable goals and ultimately protect from cardio-metabolic diseases and early death [17].

One of the key predictive factors for metabolic disruption in obesity is insulin-like growth factor 1 (IGF-1), a mitogenic hormone involved in several processes like growth, angiogenesis and differentiation. In individuals with obesity, lower IGF-1 serum levels and a blunted response to growth hormone-stimulating dynamic tests are associated with greater metabolic impairment [18,19,20,21,22,23,24,25]. However, the usefulness of IGF-1 serum measurement is limited by a poor standardization of its normal values, as they vary significantly with gender, age and body fat [26]. In order to overcome this limit, the IGF-1 z standard deviation score (IGF-1 zSDS) has been previously adopted as a surrogate marker of IGF-1 normalized by age, gender and BMI [27].

Taking these considerations into account, the aim of the study was to define a model predicting the diagnosis of MHO in the cohort of patients that have accessed the High Specialization Centre for the Care of Obesity, Sapienza University of Rome, between 2010 and 2019 through ML technology.

In particular, we aimed to:(1)Describe the cohort of patients at the time of their first access to our obesity specialisation centre with a rigorous collection of anthropometric, clinical and metabolic data.(2)Apply AI with a logic ML approach in the obese subgroup of patients to identify new parameters possibly involved mechanistically in the pathogenesis of the metabolic syndrome (either clinical, biochemical or instrumental), which could help distinguish MUO from MHO patients and define the best model capable of predicting the development of MUO, with a special focus on IGF-1 zSDS.

## 2. Materials and Methods

### 2.1. Study Design

This was an observational retrospective study. Data were derived from a database including medical records of all patients attending the High Specialization Centre for the Care of Obesity, Sapienza University of Rome, between 2001 and 2019. The study was approved by the Medical Ethical Committee of Sapienza University of Rome (ref. CE5475) and was conducted in accordance with the Declaration of Helsinki (1964) and subsequent amendments. All patients undergoing clinical examination provided written consent upon admission to our specialisation centre. Inclusion of patients in the ML analysis was regulated by the following criteria:−Inclusion criteria: age ≥18 years old and body mass index ≥30 kg/m^2^.−Exclusion criteria: (1) pregnancy or breastfeeding; (2) patients with type 1 diabetes mellitus and severe chronic liver or kidney dysfunction; (3) tobacco habit and alcohol abuse; (4) current medication with drugs that could lead to weight gain.

### 2.2. Subjects and Measurements

All clinical, anthropometric, biochemical and hormonal parameters that are routinely part of the diagnostic path that patients undertake when hospitalized in our centre were included in the database. All patients had extensive blood tests performed, such as complete blood count and a comprehensive metabolic panel, including but not limited to renal and liver function testing, serum electrolytes and additional analyses as needed.

#### 2.2.1. Anthropometric Measurements

Anthropometric parameters were obtained between 8 and 10 a.m. in fasting subjects wearing light clothing and no shoes. Body weight was obtained with the use of a balance-beam scale (Seca GmbH & Co., Hamburg, Germany). Height was rounded to the nearest 0.5 cm. Waist circumference was measured at the level of the iliac crest and hip circumference at the level of the symphysis-greater trochanter to the closest centimetre. Subsequently, the following indirect anthropometric indices were derived: body mass index (BMI) calculated as weight divided by squared height in metres (kg/m^2^); waist hip ratio (WHR) calculated as waist circumference (cm) divided by hip circumference (cm). Arterial blood pressure was measured at the right arm, with the patients in the sitting position after five minutes of rest. The average of three different measurements with a mercury sphygmomanometer was used for the analysis.

#### 2.2.2. Routine Laboratory Assessments

Blood samples were collected between 8 and 9 a.m. by venepuncture from fasting patients. Samples were then transferred to the local laboratory and handled according to the local standards of practice.

The following assays were measured: complete blood count (CBC), fasting blood glucose (FBG), insulin, total cholesterol (TC), triglyceride (TG), high-density lipoprotein cholesterol (HDL-C), low-density lipoprotein cholesterol (LDL-C), glycosylated haemoglobin (HbA1c), aspartate aminotransferase (AST), alanine aminotransferase (ALT), alkaline phosphatase (ALP), gamma-glutamyl transferase (γ GT), serum albumin, serum creatinine, direct and indirect serum bilirubin, C-reactive protein (CRP), erythrocyte sedimentation rate (ESR), serum sodium, serum potassium, serum calcium, serum phosphorus and 25-hydroxyvitamin D.

To predict insulin resistance, a homeostatic model assessment of insulin resistance (HOMA-IR) was calculated according to the following formula: HOMA-IR = (insulin (mU/l) × fasting blood glucose (mmol/l))/22.5.

#### 2.2.3. Hormonal Assessments

In accordance with the European Society of Endocrinology Clinical Guideline on the Endocrine Work-up in Obesity [28], patients were tested for secondary forms of obesity, such as hypothyroidism or hypercortisolism, as appropriate.

TSH measurements were based on a chemiluminescent immunoassay (CLIA) using ADVIA Centaur (Siemens Medical Solutions Diagnostics, Tokyo, Japan), whereas serum cortisol was measured by an immunoradiometric assay (Abbott Diagnostics, Chicago, IL, USA).

Moreover, insulin-like growth factor 1 (IGF-1) was measured in all patients presenting with signs and symptoms of adult-onset growth hormone deficiency [29]. Specifically, IGF-1 was assayed by an immunoradiometric assay, after ethanol extraction (Diagnostic System Laboratories Inc., Webster, TX, USA). The normal ranges in <23, 23–30, 30–50, 50–100-year-old patients were 195–630, 180–420, 100–415, 70–250 mg/l, respectively. Since IGF-1 serum levels strictly depend on age and gender, we calculated the SDS of IGF-1 levels according to age (zSDS) to analyse the relationships between IGF-1 levels and the other parameters. In order to obtain a z-score, we calculated the mean and S.D. of IGF-1 levels in young (<30 years), adults (30–50 years), middle-aged (50–65 years), and elderly (>65 years) women and men, as previously described [27]. zSDS is defined by the following formula: IGF-1 zSDS = (IGF-1 − mean)/S.D.

#### 2.2.4. Dual-Energy X-ray Absorptiometry

Human body composition parameters were measured with dual-energy X-ray absorptiometry (DXA) (Hologic A Inc., Bedford, MA, USA, QDR 4500W). All scans were administered by trained research technicians using standardized procedures recommended by GE-Healthcare. The instrument was calibrated daily. Whole body as well as regional body composition were assessed. Delimiters for regional analysis were determined by standard software (Hologic Inc., Marlborough, MA, USA, S/N 47168 VER. 11.2). Regions of the head, trunk, arms and legs were distinguished with the use of specific anatomic landmarks.

Therefore, for each patient, the following parameters were measured: whole-body fat mass (FM, kg and %), truncal fat mass (TFM, kg and %), appendicular fat mass (AFM), lean mass (kg). Appendicular lean mass (ALM, kg) was determined by summing lean mass measurements of the arms and legs. Fat distribution was assessed by upper body/lower body fat index, calculated as the ratio between upper body fat (head, arms and trunk fat, kg) and lower body fat (leg fat, kg) [30].

### 2.3. Characteristics of the Logic Machine Learning (LML)

ML is a subdomain of AI that “learns” inherent statistical patterns in data to make predictions about unseen data [31]. The power of this technology involves the analysis of a plethora of variables, with subsequent identification of models that stratify patients at risk, thus guiding the appropriate therapeutic strategy [3].

A specific type of ML approach is the “rule generation method”, which constructs models that are described by a set of intelligible rules, thus allowing to derive important insights about the variables included in the analysis and their relationships with the target attribute. In particular, Rulex^®®^ (Innovation Lab, Rulex Analytics, Genova, Italy), which was chosen for this analysis, is a logic machine learning (LML) original proprietary “clear box-explainable” AI algorithm. This type of algorithm, unlike “black box” AI, does not pose the problem of transparency and can be used with the objective of understanding a given phenomenon by producing sets of intelligible rules expressed in the form “*if premise*…, *then consequence*…”, where “*premise*” refers to the combination of conditions (conditional clauses) on the input variables, and “*consequence*” contains information about the target function (yes or no/presence or absence of disease) [2,32]. Therefore, the Rulex^®®^ data analysis process can be summarized in the following steps: (1) ML technology creates a model from known variables and is able to establish a ranking with the most relevant variables that explain the starting premise; (2) the model makes it explicit if there are threshold values of the most important variables previously identified; (3) the model, if used in a prediction, starting from variables of a new patient, makes it explicit why the response is yes or no.

In our study, the *premises* were the following two: (1) “the patient is metabolically healthy” and (2) “the patient is metabolically unhealthy”. Specifically, patients were considered as metabolically healthy obese if they did not show any of the features of metabolic syndrome described by the ATP III criteria on top of increased waist circumference (≥94 cm for men and ≥80 cm for women) [33], whereas they were considered as metabolically unhealthy when two or more of the features of metabolic syndrome were present. Patients taking antidiabetic, antilipidemic and antihypertensive drugs were considered to have diabetes, dyslipidaemia and hypertension, respectively.

Sample size for ML analysis was measured using the Vapnik–Chervonenkis dimension, according to which at least 500 patients per class were required.

Rulex^®®^ ML selected the most relevant variables to predict the development of MUO, starting from all those included in the database (anthropometric data, biochemical and hormonal assays, body composition by DXA) apart from blood pressure, lipid profile and glycaemic parameters that are included in the definition of metabolic syndrome itself. Two different predictive models were created with the highest accuracy, the first including IGF-1 among the variables selected and the second with IGF-1 zSDS instead of IGF-1. Given the collinearity of these two variables, it was not possible to include them together in the same model.

## 3. Results

### 3.1. Population

Our centre registered a total of 4541 hospitalizations from 2001 to 2019. Among them, 3529 patients accessing the centre in this period were diagnosed with obesity. Of these, 2824 individuals underwent only one hospitalization, while 705 more than one in different years. Only 2567 met the inclusion criteria and were included in the ML analysis. Baseline characteristics and age distribution of the study population are summarized in Table 1, broken down by metabolic status. Specifically, metabolic syndrome, diagnosed according to the ATPIII criteria [33], was significantly more prevalent among male subjects compared to their female counterparts (Table 1). Patients with MUO had significantly higher blood pressure, HOMA-IR, uric acid, TG, total cholesterol, LDL-cholesterol and upper/legs fat ratio. Intriguingly, patients with MHO had higher IGF-1 values than their counterparts with MUO (Table 1).

The calculated IGF-1 SDS was −0.86 ± 1.98 in our population, and its distribution in the overall study population, as well as in the metabolically healthy and unhealthy obese subgroups, is summarized in Figure 1A,B, respectively. It is noteworthy that it was significantly lower in the group of patients with MUO compared to the metabolically healthy counterparts (−0.6 ± 0.8 vs. −0.2 ±0.6, *p* < 0.0001, Table 1).

### 3.2. Logic Machine Learning

We considered in the ML analysis all variables in the database, except for those included in the definition of metabolic syndrome itself, in order to identify the best model for predicting the presence/absence of MHO. The machine learning system considered all the variables in the database together and not one after the other. Six modelling cycles were performed (learning set = 70% and test set = 30%) to analyse the various facets of this phenomenon.

In the model including IGF-1, the most important variables defining the outcome, starting from the most influencing to the least, were: HOMA-IR, upper/legs fat, HbA1c, RBC, age, ALT, uric acid, WBC, IGF-1, γGT. The model was predictive of the presence/absence of metabolically healthy obesity with a precision of 66.67% and 72.15%, respectively (Figure 2A). In a second model we included IGF-1 zSDS as variable in place of IGF-1. In this model, the variables defining the outcome were: HOMA-IR, HbA1c, age, upper/legs fat, RBC, ALT, WBC, γGT, uric acid, neutrophils, AST, IGF-1 zSDS. In particular, in this model IGF-1 zSDS values >0.03 and <0.52 predicted the presence/absence of MHO, respectively. Overall, the model increased its precision, reaching the value of 71.84% for the presence of MHO and 72.3% for its absence (Figure 2B).

## 4. Discussion

In the current study (1) we described the characteristics of a relatively large population of patients with obesity admitted to an Italian third tier obesity centre; (2) we adopted an ML approach to identify the variables involved in the characterization of MHO in the study population.

Notably, we found that more women than men were hospitalized for obesity in the study period. Moreover, male subjects were significantly more likely to be diagnosed with MS, hypertension, dyslipidaemia and diabetes mellitus compared to the female counterpart. This is in accordance with previous studies showing that women seek for medical attention earlier than their male counterparts and that MS prevalence is higher among men compared to women [34,35].

Moreover, we identified two models predicting the presence of MHO in our study population through the use of an ML approach, including all the anthropometric, general and biochemical data collected during hospitalisation. In both models, HOMA-IR proved to be a robust tool for the characterisation of metabolic phenotype among patients with obesity, as values >3.48 and <2.48 (in model 1) or >2.47 and <2.10 (in model 2) identified MUO and MHO patients, respectively. These results are close enough to the optimal cutoffs identified by Gayoso-Diz and colleagues, who found that HOMA-IR levels significantly increased with rising number of MS components from 1.7 (without MS components) to 5.3 (with five components) [36]. ML confirmed that insulin resistance appears to be one of the main players in the pathophysiology of metabolic derangement in obese patients, an aspect that was already emphasised in the original, but now outdated, WHO definition of MS in 1998 [37], although it is no longer a requirement to make a diagnosis.

Furthermore, a previous study showed that there are age and gender-specific differences in HOMA-IR levels, with increased levels in women older than fifty [38]. Interestingly, 50 years of age is the same threshold value identified by Rulex^®®^ to discriminate between MHO and MUO. This result provides evidence that there are age differences in the way metabolic health is expressed and that, as already proved [39], the prevalence of MS and consequently of MUO has a steep increase with age. In this regard, recent strands of research suggest that the prevalence of MUO increases with menopause and may partially explain the apparent acceleration in cardiovascular diseases after menopause [40,41], although menopause may be considered a predictor of MS independent of women’s age [42].

Although there is no doubt that insulin resistance is the major aetiological factor in the development of MS, Osei and colleagues have recently investigated the significance of HbA1c as a surrogate marker for MS, showing that in subjects with increased HbA1c, some, albeit not all, of the components of MS could be defined by HbA1c [43]. In this regard, as suggested by the Rulex^®®^ model, a glycosylated haemoglobin above 5.25%, although not diagnostic for diabetes or prediabetes, contributes to the identification of metabolic impaired patients. Our finding confirms that HbA1c may be a valid predictor of MUO status [44] and the threshold value we found reflects what is currently reported in the literature according to which a HbA1c of 5.45% can predict the presence of MS [45]. Moreover, elevated levels of serum uric acid (SUA) have been suggested to associate with cardiovascular disease, obesity and MS [46]. In this regard, the ML analysis confirmed that patients with normal levels of SUA, and specifically below 6.25 mg/dl, are more likely to have MHO.

Another interesting parameter that was identified by ML in predicting MUO is the value of liver enzymes. Specifically, ALT levels above 29.35 U/L (first model) or 28.9 U/L (second model) describe the cohort of patients with MUO. A slight increase in liver indices, especially AST, can be considered as a red flag for the development of nonalcoholic liver disease (NAFLD), commonly recognized as the hepatic manifestation of the MS, as reflected by the presence of ALT, AST and BMI in the surrogate marker of NAFLD hepatic steatosis index (HSI) [47,48]. ML confirmed that in subjects with obesity or MS, screening for NAFLD by liver enzymes and/or ultrasound should be part of routine workup, as recommended in the clinical practice guidelines for the management of NAFLD provided by the European Association for the Study of Obesity [49]. ML also proved that ALT values in the normal range may play a role in the identification of MHO patients, but failed to define a specific threshold value for ALT in predicting MUO. Regarding γGT, which was also included in the models, serum levels higher than 17.45 U/L (first model) or 11.1 U/L (second model) identify the group of patients with MUO. Of interest, both AST and γGT are already included in validated, noninvasive tools for the assessment of liver fibrosis such as Fibrosis-4 (FIB-4), NFS (NAFLD Fibrosis Score) and fatty liver index (FLI) [50]. In light of this, as recently suggested by Godoy-Matos et al., the proper understanding of NAFLD spectrum—as a continuum from obesity to MS and diabetes—may contribute to the early detection and to the establishment of a targeted treatment [47,51].

Among all the variables of fat distribution evaluated with DXA, the upper/leg fat index was identified by ML as the best predictor of MUO. An elevated ratio (>2.01), as reported in our analysis, indicates upper body fat accumulation and central obesity, which both lead to metabolic complications; contrarily to lower body fat, which confers reduced risk [52]. Additionally, as we have already described, prominent upper body fat deposition is likely to predispose individuals to apnoea. Indeed, fat accumulation in strategic locations, such as the head and upper airway, predisposes to pharyngeal narrowing and upper airways collapsibility resulting in obstructive sleep apnoea syndrome (OSAS) [30]. In turn, OSAS is a risk factor for insulin resistance and diabetes and is often found in the setting of MS. Occasionally, in a subset of patients with OSAS, secondary polycythaemia will develop [53].

Even though a true polycythaemia is not generally found, according to our analysis an RBC count >4.45 (10^12^/L) is a predisposing factor for MUO. When exclusively considering the female population, the calculated cutoff was higher (>4.74 10^12^/L). These results are along the line of already published data reporting that subjects affected by MS exhibit a higher count of RBCs compared to metabolically healthy subjects. It has been reported that, despite the presence of chronic inflammation which has suppressive erythropoietic effects, erythropoiesis correlates with central obesity and insulin resistance [54] and that RBC count is, even though still within normal range, significantly higher in the presence of MS for each sex [55].

Innumerable etiopathogenetic mechanisms responsible for the onset of MS among patients with obesity have been identified, but chronic, low-grade and systemic inflammation has been acknowledged as the common denominator [56]. The WBC count is an objective marker of acute infection, tissue damage and inflammation [57]. A few studies have already confirmed that the WBC count is correlated with the increase of certain variables of MS [58]. In this regard, our analysis found that a neutrophilic leucocytosis is often common in MUO, suggesting an altered immune response and increased susceptibility to bacterial and viral infections, as known from the recent COVID-19 pandemic [59,60,61,62] and previous cross-sectional studies [63].

A further key predictive factor in the development of MS is IGF-1, a polypeptide hormone structurally similar to insulin, which promotes tissue growth and maturation through upregulation of anabolic processes. Adult-onset growth hormone deficiency (GHD) is relatively common in patients with obesity, being associated with a worse metabolic profile [64,65]. Epidemiological studies have suggested that IGF-1 levels in the upper normal range are associated with increased insulin sensitivity, better liver status and reduced blood pressure [66,67,68,69].

Noteworthy, the first model provided by Rulex^®®^ including IGF-1, was predictive of the presence/absence of metabolically healthy obesity with a precision of 66.67% and 72.15%, respectively. However, the usefulness of IGF-1 serum measurement is limited by a poor standardization of its normal values, as both age and gender can significantly affect serum IGF-1 concentrations. By the age of 65 years old, daily spontaneous GH secretion is reduced by up to 50–70%, and consequently IGF-1 levels decline progressively as they vary significantly with gender, age and body fat, similar to what happens with bone mineral density (BMD). This leads to the need of a score keeping these factors into consideration, such as the T- and Z-score developed to better evaluate BMD. In this regard, when added IGF-1 zSDS as a variable, our second model increased its precision, reaching the value of 71.84% for the presence of metabolically healthy obesity and 72.3% for its absence.

Our study suggests that ML may have a broad application in the risk stratification of people suffering from obesity and supports its potential role in the health care system to identify those at higher risk, among the wide population of subjects with obesity, and to identify the parameters characterising the state of MHO, a phenotype that could represent the first goal to be achieved in the management of chronic obesity in order to reduce the risk of death. Moreover, we found that the surrogate marker IGF-1 zSDS, more than IGF-1 alone, can increase the precision of the model in the prediction of the presence/absence of MHO, suggesting its potential application in clinical practice as a marker of metabolic impairment.

The strengths and limitations of this study warrant mention. Firstly, this study was conducted in a large cohort that was nationally representative of the Italian obese population. However, our patient cohort is not gender balanced. The main limitation of the study is that Rulex^®®^, like many other ML algorithms, needs a large amount of data to yield relevant results. Further prospective studies, with a larger number of patients, and comparison studies with other supervised machine learning models, such as support vector machine, naïve Bayes algorithm and random forest algorithm, are needed to confirm our results.

## 5. Conclusions

Integration of ML technology in medicine may help scientists understand in a deeper way the pathogenesis of complex diseases, such as the metabolic ones. One possible application of this ML analysis is the development of an algorithm, which, in a similar way to the fracture risk assessment tool (FRAX) for osteoporosis [70], can accurately predict the risk of developing MUO at 5 or 10 years in the population of patients with obesity, thus identifying the clinical phenotype with the highest risk and encouraging more and more precise and targeted therapeutic approaches.

## Figures and Tables

**Figure 1 nutrients-14-00373-f001:**
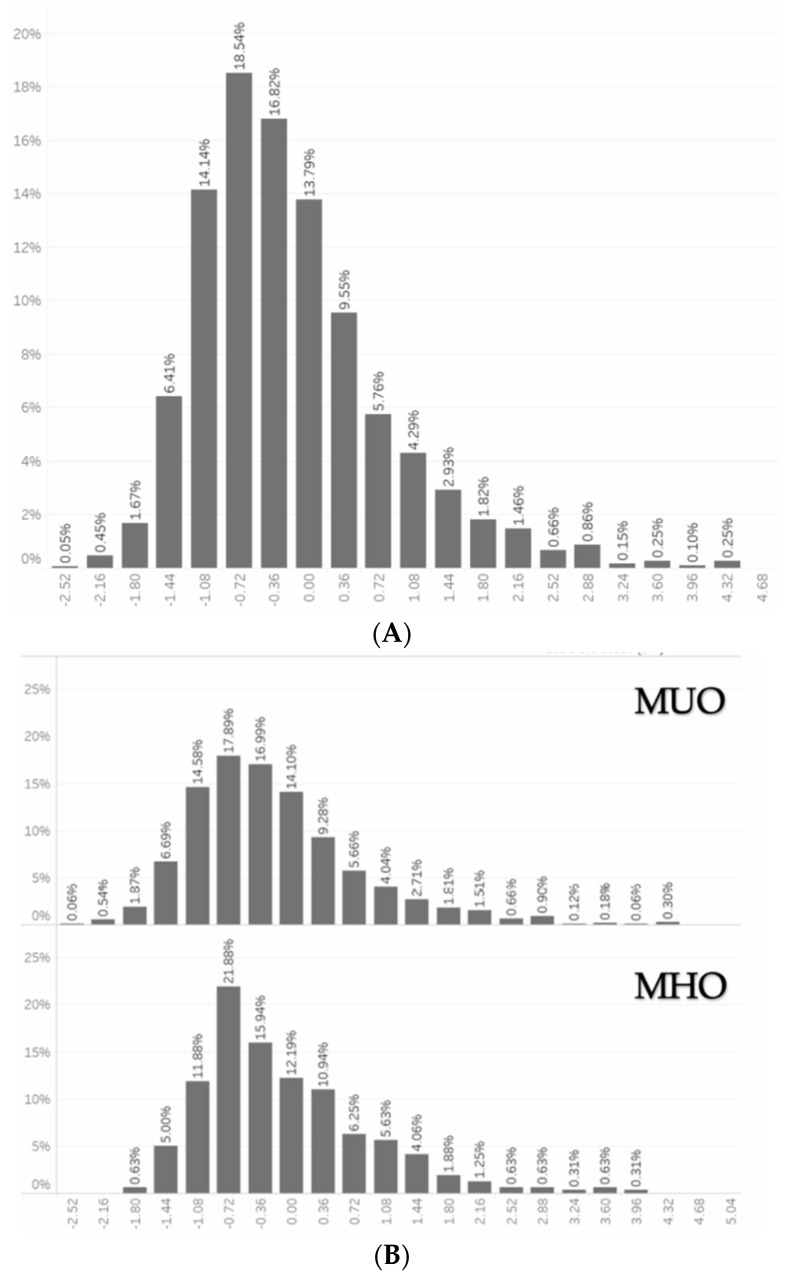
(**A**) Distribution of IGF-1 zSDS in the overall study population. (**B**). Distribution of IGF-1 zSDS in the MUO and MHO subgroups. Abbreviations: IGF-1 zSDS, insulin-like growth factor 1 z standard deviation score; MUO, metabolically unhealthy obese group; MHO, metabolically healthy obese group. Variables are expressed as percentile of total population.

**Figure 2 nutrients-14-00373-f002:**
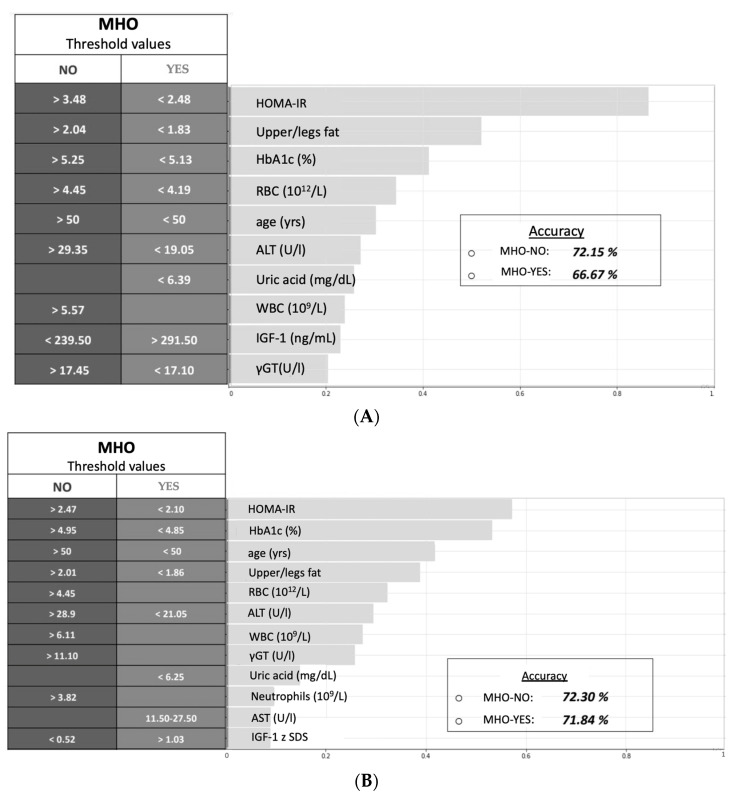
(**A**) Model no. 1 with the most relevant variables and threshold values that predict the development of MUO. (**B**) Model no. 2 with the most relevant variables and threshold values that predict the development of MUO. Abbreviations: yrs, years; HOMA-IR, model assessment of insulin resistance; HbA1c, haemoglobin A1C; RBC, red blood cell; ALT, alanine aminotransferase; WBC, white blood cell; γGT, gamma-glutamyl transferase; AST, aspartate aminotransferase, IGF-1 zSDS, insulin-like growth factor 1 z standard deviation score; MUO, metabolically unhealthy obese group; MHO, metabolically healthy obese group. IGF-1, insulin-like growth factor 1.

**Table 1 nutrients-14-00373-t001:** Baseline characteristics of study population included in the ML analysis, broken down by presence/absence of metabolic impairment.

	MHO (*n* = 695)	MUO (*n* = 1872)	Overall (*n* = 2567)
Age (yrs)	45.9 ± 13.5	47.6 ± 13.5 **	47.1 ± 13.4
Gender (%F)	82.3%	74.6% *	76.7%
Obesity duration (yrs)	25.5 ± 15.4	26.4 ± 15.1	26.1 ± 15.2
BMI (kg/m^2^)	38.0 ± 6.1	39.8 ± 6.8 ***	39.3 ± 6.6
WC (cm)	116.6 ± 15.3	121.9 ± 15.4 **	120.5 ± 15.4
HC (cm)	121.5 ± 14.5	122.4 ± 14.9	122.2 ± 14.7
WHR	0.95 ± 0.12	0.99 ± 0.09	1.0 ± 0.1
SBP (mmHg)	126.4 ± 10.9	131.9 ± 16.3 *	130.4 ± 15.2
DBP (mmHg)	79.3 ± 10.8	83.1 ± 11.1 **	82.1 ± 11.0
IGF-1 (ng/mL)	165.2 ± 77.2	154.4 ± 74.5 *	157.3 ± 76.1
IGF-1 zSDS	−0.96 ± 2.3	−1.1 ± 1.96	−1.1 ± 2.1
AST (U/L)	19.5 ± 7.5	22.1 ± 12.1 ***	21.4 ± 8.7
ALT (U/L)	23.7 ± 16.4	30.3 ± 22.1 ***	28.5 ± 21.3
γ GT (U/L)	23.4 ± 24.4	28.9 ± 16.5 *	27.4 ± 19.4
Uric acid (mg/dL)	4.9 ± 1.3	5.5 ± 1.5 ***	5.3 ± 1.4
HOMA-IR	3.5 ± 3.2	5.7 ± 5.4 ***	5.1 ± 4.5
HbA1c (%)	5.7 ± 1.1	6.2 ± 1.1	6.1 ± 1.1
Vitamin D (ng/mL)	21.9 ± 10.2	20.5 ± 10.3 **	20.9 ± 10.3
Folate (ng/mL)	7.9 ± 23.2	8.8 ± 35.3	8.6 ± 28.4
TG (mg/dL)	91.6 ± 27.2	150 ± 80.1 ***	134.2 ± 62.7
TC (mg/dL)	144 ± 33.3	195.1 ± 41 ***	181,3 ± 37.2
HDLC (mg/dL)	59.6 ± 11.3	45.2 ± 10.6 **	49.1 ± 10.9
LDLC (mg/dL)	116.5 ± 30.7	120.1 ± 30.2 **	119.1 ± 30.5
Creatinine (mg/dL)	0.7 ± 0.16	0.8 ± 0.23	0.8 ± 0.19
Ca (mg/dL)	9.32 ± 0.44	9.34 ± 0.44	9.3 ± 0.44
Ph (mg/dL)	3.5 ± 0.5	3.5 ± 0.6	3.5 ± 0.6
Na (mmol/L)	141.5 ± 2.6	140.9 ± 2.5	141.1 ± 2.5
K (mmol/L)	4.2 ± 0.3	4.2 ± 0.4	4.2 ± 0.4
Albumin (g/dL)	4.3 ± 0.4	4.3 ± 0.4	4.3 ± 0.4
CRP (µg/L)	0.5 ± 0.5	0.7 ± 0.6 **	0.6 ± 0.6
ESR (mm/h)	26.1 ± 16.4	27.9 ± 17.2 *	27.4 ± 16.8
Body fat (%)	41.6 ± 6.3	40.7 ± 6.7 **	40.9 ± 6.5
Lean mass (%)	58.4 ± 6.4	59.3 ± 6.7 **	59.1 ± 6.6
Trunk fat (%)	39.1 ± 6.5	39.4 ± 6.5	39.3 ± 6.5
Upper/legs fat	1.62 ± 0.3	1.97 ± 0.36 ***	1.9 ± 0.32

Abbreviation: MHO, metabolically healthy obese; MUO, metabolically unhealthy obese; yrs, years; BMI, body mass index; WC, waist circumference; HC, hip circumference; WHR, waist to hip ratio; SBP, systolic blood pressure; DBP, diastolic blood pressure; IGF-1, insulin-like growth factor 1; IGF-1 zSDS, insulin-like growth factor z standard deviation score; AST, aspartate aminotransferase; ALT, alanine aminotransferase; γ GT, gamma-glutamyl transferase; HOMA-IR, model assessment-estimated insulin resistance; HbA1c, haemoglobin A1C; TG, triglycerides; TC, total cholesterol; HDLC, high-density lipoprotein cholesterol; LDLC, low-density lipoprotein cholesterol; Ca, calcium; Ph, phosphate; Na, sodium; K, potassium; CRP, C-reactive protein; ESR, erythrocyte sedimentation rate. * *p* < 0.05. ** *p* < 0.01. *** *p* < 0.001.

## Data Availability

Data will be made available upon reasonable request to the corresponding author.

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
