# Peer review of "Application of a Machine Learning Technology in the Definition of Metabolically Healthy and Unhealthy Status: A Retrospective Study of 2567 Subjects Suffering from Obesity with or without Metabolic Syndrome"

_nutrients, 2022, doi:10.3390/nu14020373_

Round 1
Reviewer 1 Report
Authors of the manuscript entitled „Application of a Machine Learning Technology in the Definition of Metabolically Healthy and Unhealthy Status Among Patients Suffering from Obesity: a retrospective study of 2567 subjects” presents an interesting topic in the field of excess body weight, but major corrections are necessary.
Introduction
- In the Introduction, Authors do not mention the metabolic syndrome despite the fact that, based on the elements of the metabolic syndrome, they classified patients into healthy and metabolically unhealthy. It is necessary to modify the introduction and title of this manuscript.
- Did Authors exclude from the study people with diseases or medications that may cause weight gain?
- The aim of the study should be at the end of the introduction, not as a separate point.
Materials and methods
- How was the sample for the study calculated?
- Please complete the description of the methodology of biochemical determinations.
Results
- The average BMI in women indicates overweight, not obesity. Please indicate the% of overweight and obese patients of 1st, 2nd and 3rd degree.
- Figure 2a and 2 b, Figure 3 a and 3 b are illegible.
- The meaning of this study is not fully understood - there are simple possibilities of assessing whether a patient has metabolic syndrome or not, so what might be the usefulness of this study?
Reviewer 2 Report
The overall aim of study is very interesting that IGF-1 as a novel metabolic health parameter to effectively predict the development of metabolically unhealthy (MUO) obesity using ML technology. I do however have some major concerns regarding the machine learning approach and on the reusability of their process. Major concerns are listed below:
Comment 1: This paper uses several data analysis methods including machine learning. The authors should clarify that all the methods are open to the readers using a summary table and source data & codes should be made available through a repository such as Github.
Comment 2: How does a ML model perform when the IGF-1 factors are used as inputs?
Comment 3: I found all Figure quite difficult to parse. I don't know if it's the colour scheme or just my eyes, but I had to squint quite hard to figure out. Particularly, Figure 2. A and Figure 2. B.
Comment 4: The authors claim that IGF-1 was selected based on performance, but I could not see a comparative graph or table showing the results of the various approaches they tested.
Comment 5: Why did the authors not fit a single ML model, instead of having two independent models?
Comment 6: Following Comment 5, the authors should compare with other ML models like random forest, SVM, etc, so as to better claim the advantage of their method.
Comment 7: The authors declare that “ML probes information without prior knowledge and assumptions”. This statement is incorrect, as some ML models need to define prior distribution (like Bayesian models), while some only work under concern conditions/assumptions (like PCA cannot fully disentangle those nonlinear manifolds). In fact, all ML models have priors and work under certain conditions. Even the most powerful deep models need to assign the structure information in advance before the proper training procedure.
Round 2
Reviewer 1 Report
- Figures are still illegible.
- Authors should refer to the usefulness of ML technology, because its usefulness is still not very clear.
Reviewer 2 Report
The authors answered all my questions.